# MyPal-Child study protocol: an observational prospective clinical feasibility study of the MyPal ePRO-based early palliative care digital system in paediatric oncology patients

Marcel Meyerheim  ,[1] Christina Karamanidou,[2] Sheila Payne,[3] Tina Garani-Papadatos,[4] Annette Sander,[5] Julia Downing,[6] Kostas Stamatopoulos,[2] Julie Ling,[7] Cathy Payne,[7] Lydia Scarfò ,[8] Petr Lokaj,[9] Christos Maramis,[2] Norbert Graf [1]

For numbered affiliations see end of article.

**Correspondence to**
Professor Norbert Graf;
graf@uks.eu

## ABSTRACT

**Introduction** Electronic patient-reported outcomes (ePROs) have tremendous potential to optimise palliative and supportive care for children with cancer, their families and healthcare providers. Particularly, these children and their families are subjected to multiple strains caused by the disease and its treatment. The MyPal digital health platform is designed to address these complex demands by offering pursuant ePRO-based functionalities via two mobile applications, one developed for children and the other for their parents.

**Methods and analysis** In this observational prospective feasibility study, 100 paediatric oncology patients aged between 6 and 17 years and at least one of their parents/legal guardians will be recruited at three clinical sites in two European countries (Germany and Czech Republic). They will use the mobile applications which are part of the novel digital health platform. During a 6-month study period, participants will complete various ePROs via the applications addressing quality of life, satisfaction with care and impact of the disease on the family at monthly intervals. Additionally, priority-based symptom reporting is integrated into a serious game for children. Outcomes that will be assessed concern the feasibility and the evaluation of the newly designed digital health platform to contribute to the evidence base of clinical ePRO use in paediatric oncology and palliative care process.

**Ethics and dissemination** The MyPal-Child study obtained ethical approval from the Ethics Committee responsible for the University of Saarland, that is, the Ärztekammer des Saarlandes, the Ethics Committee of the Medical School Hannover and the Ethics Committee of the University of Brno. Study results will be disseminated through scientific publications, presentations at international conferences, congresses and a final report to the European Commission. General publicly accessible information can be found on the project website (www.mypal-project.eu) and social media.

**Trial registration numbers** U1111-1251-0043, DRKS00021458, NCT04381221.

## Strengths and limitations of this study

► Multicentre (three clinical sites in two European countries) observational study on the feasibility and acceptability of using electronic patient-reported outcome (ePRO)-based systems in palliative care for paediatric oncology patients.

► Patient outcomes are reported independently in an inpatient/outpatient setting and the impact of this approach will be evaluated by the treating healthcare professionals.

► The design of the study is the product of a joint effort of interdisciplinary expertise from clinicians, palliative care experts, software engineers and scientists.

► The regular review of study participants' reportings demands healthcare professionals' additional time. The investigation of this factor is considered as a secondary objective in the elucidated study.

► Future randomised controlled trials are required to investigate the effectiveness of ePROs in paediatric palliative care management and treatment.

## INTRODUCTION

According to WHO definition, paediatric palliative care for children and their families is defined as 'the active total care of the child's body, mind and spirit, and also involves giving support to the family. It begins when illness is diagnosed, and continues regardless of whether or not a child receives treatment directed at the disease. Health providers must evaluate and alleviate a child's physical, psychological and social distress. Effective palliative care requires a broad multidisciplinary approach that includes the family and makes use of available community resources; it can be successfully implemented even if resources are limited. It can be provided in

tertiary care facilities, in community health centres and even in children's homes'.[1] Unsurprisingly, this definition applies to children with cancer who require complex multimodal anticancer treatments but also benefit from palliative care supporting both the patients and their families to manage symptoms of the disease, side effects of treatment as well as treatment-related toxicity. Palliative care is not limited to end-of-life treatment but preferably starts close to the time of diagnosis and continues during as well as outside the acute disease treatment phase to improve the quality of life (QoL). Such an early integration of palliative care either along with anticancer treatment or even as the only means of treatment in rare cases, can improve both patient and caregiver outcomes.[2] The latter is particularly relevant for parents and siblings of children or adolescents with cancer who are also impacted by the disease, its treatment as well as its prognostic uncertainty.

As children with cancer and their carers often believe that the adverse effects of anticancer treatment are inevitable, they often downplay symptom severity during clinical interviews,[3] it is therefore possible that more accurate assessments can optimise symptom reporting and generate richer and more reliable data. In that respect, major improvements can be anticipated through standardising patient-reported outcomes (PROs), defined as 'measurements based on a report that comes directly from the patient about the status of a patient's health condition without amendment or interpretation of the patient's response by a physician or anyone else'.[4 5]

Overall, the use of PROs has become a prominent topic in healthcare innovation, highlighting the role of the patients' experience of their status as a key measure of healthcare quality, especially in cancer.[6] PROs provide a more comprehensive view on the patient's experience outside the environment of care and complement data gathered by Health Care Professionals (HCPs) or physicians. On these grounds, solutions are evidently required for facilitating the use of PROs in routine clinical practice of paediatric oncology and palliative care,[7 8] and these can be offered, among others, by information and communication technologies (ICT). Published evidence also supports the incorporation of electronic PRO (ePRO) assessments into healthcare as these have the potential to improve the quality of care delivered to patients with cancer.[9]

MyPal is a Horizon 2020 Research and Innovation Action aiming to foster palliative care for people with cancer by leveraging ePRO systems through their adaptation to the personal needs of the person with cancer and his/her caregiver(s). It aspires to empower people with cancer and their carers in capturing more accurately their symptoms/conditions, communicate them in a seamless and effective way to their healthcare providers and, ultimately, foster action through advanced methods of identification of important deviations relevant to the patient's state and QoL. Providing this information in a timely and comprehensive manner throughout the

disease course will reinforce the potential of applying a patient-centred and integrated palliative care approach for cancer patients with the participation of all relevant healthcare professionals (ie, oncologists, specialised physicians, psychologists, nurses) to cope with the specific disease.

In order to accomplish its mission, MyPal will exploit technological advances in ICT for supporting patients, family members and healthcare providers through a systematic and comprehensive ePRO-based digital health platform. It will demonstrate and validate the proposed add-ons to care for two different patient groups, that is, adults suffering from haematological malignancies and children with solid tumours or leukaemia, hence targeting different age groups and cancer types, through two carefully designed clinical studies that will be conducted in diverse healthcare settings across Europe. The observational study with children with cancer (namely MyPal4Kids) is the subject of the presented protocol.

## METHODS AND ANALYSIS
### Study design
This is a multinational non-experimental observational prospective feasibility study, enrolling paediatric oncology patients as well as at least one of their parents or legal guardians at three clinical sites in Europe. The emphasis is on data collection from the study participants during their 6 months of study enrolment via the functionalities of the newly developed MyPal digital health platform. The methodological approach involves two main mobile applications (see the section MyPal Digital Health Platform). The first comprises of a so-called 'serious game' developed with the aim to combine the advantages of modern ePROs with the motivational aspects of gaming by wrapping symptom reporting in the entertaining cover of an age-appropriate, non-violent game. Such an approach of gamification follows recommendations to encourage engagement of young patients and to split up otherwise long assessment questionnaires.[10]

The second concerns convenient self-reporting and proxy reporting for carers, that is, in the context of this study the child's parents/legal guardians and healthcare providers, via ePROs as outcome measures. The study aims to evaluate the feasibility of integrating the developed ePRO-based system for palliative care in children with cancer into paediatric oncology care and assessing its benefit and support for the child and their parents. Study participant recruitment has started in December 2020 with an estimated study completion in March 2022.

### Objectives and outcome measures
#### Primary objective
The main objective is to assess the feasibility and acceptability of a comprehensive, patient-centred service for palliative care in peadiatric oncology by adapting and advancing ePRO systems.

The corresponding primary outcome measures are the rates of recruitment, participation, adherence and premature discontinuation to the different components of the MyPal digital health platform during study enrolment. Further primary outcome measures are quantitative and qualitative data to be collected as follow-up to study enrolment: quantitative data from parents through the standardised System Usability Scale (SUS)[11] as well as from the children through a newly adapted version of the SUS; qualitative data through structured interviews and focus groups with participants to identify further barriers, facilitators, preferences and engagement with regards to the MyPal digital health platform as well as differences between the participating clinical sites.

## Secondary objectives

Secondary objectives of this study are to demonstrate the appropriateness and acceptability of several ePRO assessment tools, specifically:

► Of children's symptom burden through a novel digital adaptation of the validated print versions of the Mini-SSPedi and SSPedi questionnaires[12 13] as secondary outcome measure during study enrolment.
► Of children's QoL under cancer treatment through the Pediatric Quality of Life Inventory (PedsQL) Cancer Module[14 15] as a secondary outcome measure during study enrolment.
► Of parents' burden through the Impact on Family Scale as a secondary outcome measure during study enrolment, specifically with regards to: financial impact, family-social impact and personal strain for the primary family carer.[16]
► Of the QoL of parents having a child with cancer to be assessed through the European Quality of Life 5 Dimensions 3 Level Version (EQ-5D-3L)[17] as a secondary outcome measure during study enrolment. The parents' QoL will be evaluated with regards to the following dimensions: mobility, self-care, usual activities, pain/discomfort and anxiety/depression.
► Of parents' satisfaction with the care offered to their children with cancer using the European Organisation for Research and Treatment of Cancer satisfaction with cancer care core questionnaire (EORTC PATSAT C33)[18 19] as a secondary outcome measure during study enrolment, which assesses patients' perception of the quality of medical and nursing care, as well as the organisation of care and services of an oncology department. It has been adapted appropriately to assess parents' satisfaction with children's cancer care in agreement with the authors of the questionnaire. Further secondary objectives are:
► To determine the usage and evaluation of the MyPal apps including the gamified ePRO by children with cancer. This objective is linked to both the primary outcome measures as well as to EORTC PATSAT C33 as secondary outcome measure.
► To determine the impact and effect on healthcare professionals in two European countries due to the

---

**Box 1   Eligibility criteria**

**Inclusion criteria for children**
► 6–17 years of age.
► Diagnosed with paediatric leukaemia or solid cancer in the past 12 months.
► Receiving anticancer treatment at one of the participating clinical sites.
► Have age-appropriate speaking, reading and comprehension skills in either the German or the Czech language.
► Access to an internet connection and mobile device (eg, smartphone or tablet).

**Inclusion criteria for parents**
► Parent(s) with a child eligible for the study, as per the inclusion and exclusion criteria.
► Ability to speak, read and understand German or Czech language.
► Access to an internet connection and mobile device (eg, smartphone or tablet).

**Exclusion criteria for children and parents**
► Anyone who is not able to participate in the study according to the clinical judgement of the site chief investigator or any other authorised person of the research team. This judgement has to be documented for each child/parent not being enrolled.

---

integration of ePROs in palliative care. The corresponding secondary outcome measure is a specifically developed web-based online questionnaire to be completed as follow-up to study enrolment, evaluating strain parameters such as additional time spent on care, usability, user-experience.
► To contribute to the evidence-base of the effectiveness of ePROs in palliative care and paediatric oncology which is linked to Mini-SSPedi/SSPedi and to the developed web-based online questionnaire as secondary outcome measures.

## Patient recruitment

In total, 100 children with leukaemia or solid tumours and their parents will be prospectively enrolled in the study across all three participating clinical sites:
1. University of Saarland (Germany).
2. Hannover Medical School (Germany).
3. University Hospital Brno (Czech Republic).

At each clinical study site, participants will be screened for eligibility and contacted by the local study team (see box 1 for eligibility criteria). Information sheets will be handed to the parents and their child (see online supplemental appendix 1). For each eligible participant, informed consent will be sought prior to any study-related activities (see online supplemental appendix 2). As far as paediatric patients are concerned, information sheets and consent forms have been designed according to different age groups included in the study. Parents and children will be given an appropriate time of at least 24 hours to consider participation and to ask follow-up questions.

The patients' and parents' written consent constitutes the start of study enrolment. During the 6 months of study enrolment, children will be asked to complete questionnaires implemented as ePROs and integrated into

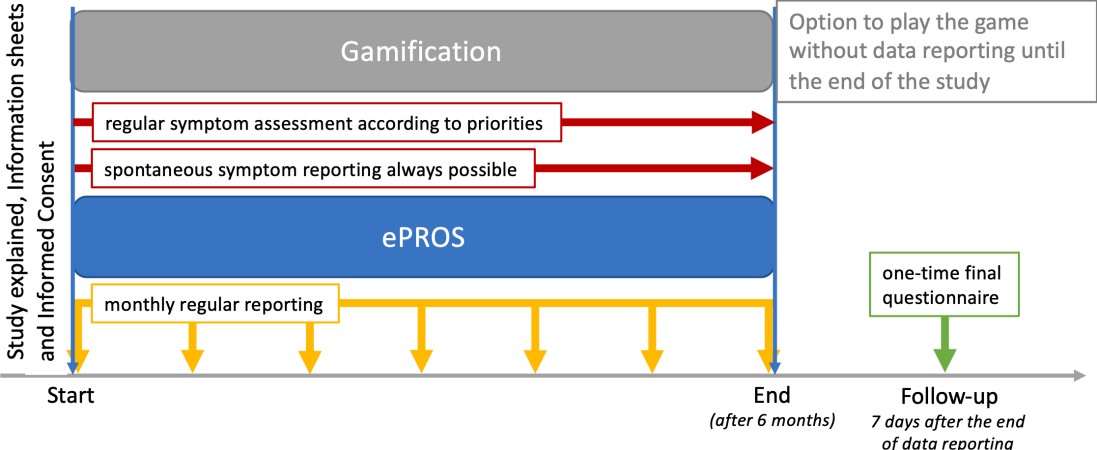

**Figure 1** Patient-specific study enrolment schema of MyPal4Kids. ePROS, electronic patient-reported outcomes.

the serious game via the MyPal platform both at baseline and at regular time intervals after receiving an introductory session by a research team member and in-app tutorials (see figure 1). Parents as well as HCPs can help the children and complete the questionnaires as proxies throughout the study. Parents will also complete periodic questionnaires at baseline and subsequently using an app provided by the MyPal digital health platform. Independently, the medical care of the child will continue as required throughout and after the course of the study, for example, anticancer treatment, receiving palliative and/or supportive care. After 6 months of study enrolment, children will be offered to continue playing the game without further data reporting or questionnaires via the MyPal digital health platform. Likewise, data will no longer be collected from their parents. Seven days after the end of data reporting, paper-based follow-up surveys will be completed by the children and their parents to evaluate their judgement on the MyPal platform and its usability.

HCPs will be asked to evaluate the impact and usability of the MyPal platform by a web-based online questionnaire to be completed at the end of the study or earlier if circumstances, for example, leaving a job, suggest it.

### MyPal digital health platform

The MyPal digital health platform and the usage of its applications is the central innovation introduced and tested for feasibility and acceptability in this study. A participatory design was adopted in the development of the digital health platform involving a series of focus

groups and discussions with patients and parents in order to identify needs and preferences as well as to validate tools, assess user experience. The usage of the MyPal platform revolves around the reporting of physical symptoms caused by the disease and medication as well as QoL of the child with cancer via the MyPal Child app. Secondarily, it comprises the reporting of their parents or legal guardians via the MyPal Carer app. The types of users of the MyPal digital health platform are specified in table 1 while the software and hardware modules of the system are presented in figure 2.

Other parameters (such as satisfaction with care, impact of disease on family, parents' QoL) are also reported by the parents via the MyPal Carer app. The symptom-related information is delivered instantly to the account of the treating HCP and can be visualised within a MyPal web-based application. The usage of the MyPal functionalities is described in more detail below from the standpoint of the patient, the parent and the HCPs.

### Children's involvement

The game is designed for a target group from 6 to 17 years old, requiring basic reading skills. Voice-over is offered to support younger children. The art style is defined by a colourful underwater exploration (see figure 3), and specifically designed to appeal to a wide age range. Designed as a runner game, the game character is continuously running through levels of a game environment with collectable items and obstacles while being controlled by the player. Runs are paused to answer single symptom-related questions inside the game (see table 2). Factors

| Table 1 | The primary and secondary users of the MyPal digital health platform | |
|---|---|
| **User type** | **Description** |
| Children with cancer | The child cancer patients registered at the participating clinical centres (primary users) |
| Parents of children with cancer | The parents or legal guardians of the primary users (secondary users) |
| Healthcare professionals | The interdisciplinary team of treating clinicians (oncologists, haematologists) nurses, psychologists, social workers, other palliative care members) of participating clinical centres (secondary users) |

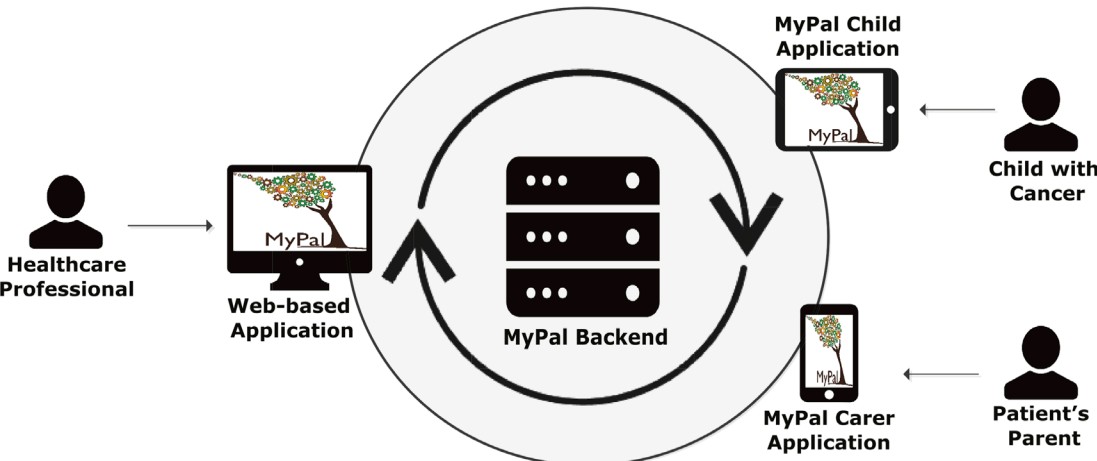

**Figure 2** Software and hardware modules of the MyPal digital health platform. The MyPal Child and carer APP are useable via various types of mobile devices. The user of the MyPal carer APP are the parents/legal guardians involved in the study. Healthcare professionals can use the MyPal carer APP as well to report as proxy for the child.

for long-term and short-term motivation, for example, collectibles and rewards, high scores and customisable game character have been incorporated into the game to make the regular use of the app and the symptom assessment as an integrated ePRO appealing and entertaining over a long period of time.

### Registration phase

Registration requires actions both in the MyPal Child app and the MyPal web-based app by the HCP. During this phase (1) the patient is registered into the MyPal digital health platform by an HCP or research team member, for example, research nurse, who also helps with the initial login; (2) a number of preferences are set; (3) baseline assessment of the patient's physical symptoms are collected. In the MyPal Child app, the patient receives a

short training session and an in-app tutorial helps to get familiar with the functionalities.

### Main usage phase

The main usage phase lasts for 6 months, during which the patient is given access to all the functionalities of the MyPal Child app, outlined in table 2. The first three functionalities can be accessed by the patient at any point in time, while the last is activated/deactivated automatically within the MyPal digital health platform on a monthly basis.

### Parent's involvement

The parents of children with cancer interact with the MyPal reporting smartphone app in two sequential phases, which are outlined below.

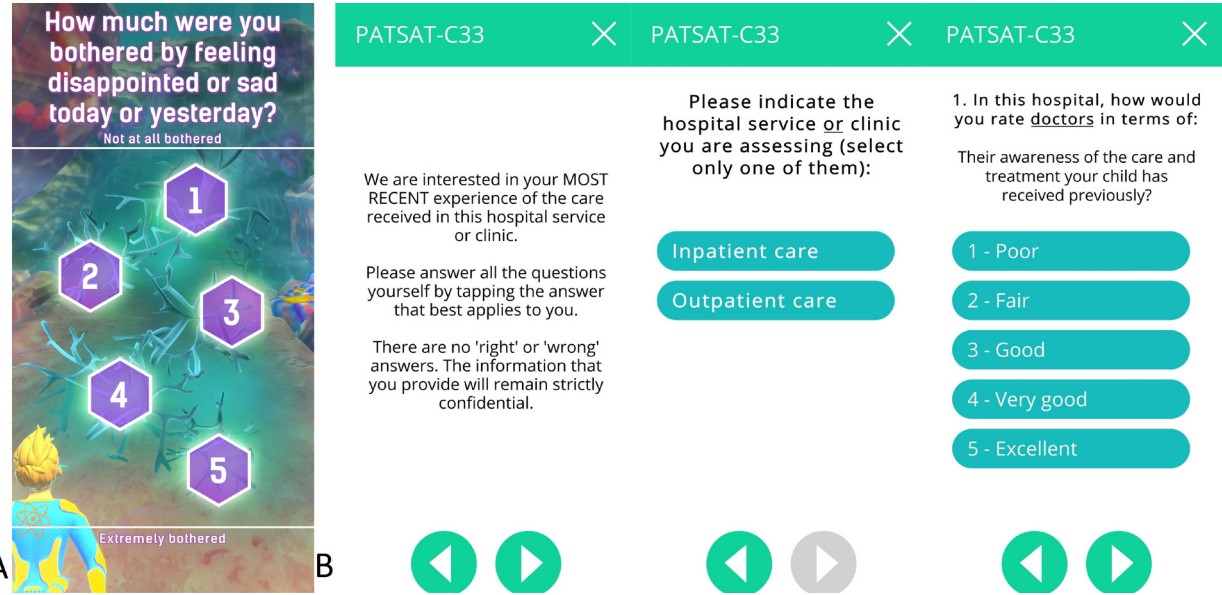

**Figure 3** (A) A screenshot of the gamified symptom questionnaire as part of the MyPal Child app. (B) Screenshots taken of the MyPal carer APP depicting the PATSAT C-33 questionnaire with instruction screen, selection of patient setting screen and one sample question screen.

**Table 2** The functionalities of the MyPal Child app offered to the child patients

| | Functionality | Description |
|---|---|---|
| P1 | Game playing | This is a prominent functionality of the MyPal gamified app. The patient is offered to play three new run sessions in a day, during which he/she controls a diver swimming in an underwater world. The patient can also customise various elements of the game (eg, the appearance of the diver) at any time. |
| P2 | In-game symptom reporting | This is the main ePRO data collection functionality of the MyPal gamified tablet app. During each gaming session, the game pauses five times for the user to answer a single symptom-related question (see figure 3); each question is seamlessly integrated in the flow of the game (the patient still sees the diver in his/her underwater environment). The set of questions to be asked stem from the unique digital adaption based on the validated SSPedi/Mini SSPedi paper-version questionnaire. Each question to be asked during a game session is determined by an internally validated novel question prioritisation algorithm. |
| P3 | Spontaneous symptom reporting | This is the secondary functionality that allows patients to report physical symptoms whenever they wish. An electronic implementation of the SSPedi/Mini SSPedi questionnaire outside the game is employed for the spontaneous reporting. |
| P4 | Periodic QoL reporting | This is the secondary functionality that asks patients to report their QoL once per month. An electronic implementation of the validated PedsQL questionnaire is, therefore, employed. |

## Registration phase

During this phase (1) the parent is linked to their child within the MyPal digital health platform; (2) baseline assessment of a subset of the outcome measures of the study via self-reporting are collected. Parents undergo a short training session and are guided during registration by an HCP participating in the study, for example, research nurse.

## Main usage phase

The main usage phase lasts for 6 months and is aligned with the patient participation in the study. The only functionality that is offered to the parent is reporting, either as a proxy or for themselves, concerning a subset of the outcome measures of the study. The reporting functionality of the MyPal Carer app is outlined in table 3.

## HCP's involvement
### Registration phase

After a short registration procedure that takes place during their first visit to the web-based app, the HCPs can continuously interact with it throughout the duration of the study. HCPs are given a short in-app training session.

### Main usage phase

During the main usage of the MyPal web-based app, the HCPs can access symptom related data being stored in the system backend which had been collected by (1) the MyPal Child app and (2) the MyPal Carer app. HCPs are authorised to solely access the data of the patients of their associated clinical centre. The individual data of participating patients are reviewed by the associated HCP at least once every 72 hours to check for new entries. The data review and any action related to this are recorded through the web interface. The functionalities of the

**Table 3** The functionalities of the MyPal Carer app offered to the child's parents

| | Functionality | Description |
|---|---|---|
| R1 | Patient QoL reporting | Proxy reporting of the QoL for the patient using an electronic implementation of the validated PedsQL questionnaire. Older patients will be asked to complete the questionnaire directly in the MyPal Child app. This is linked with an outcome measure of the study and is reported once per month. |
| R2 | Parent QoL reporting | Reporting of the QoL of the parent using an electronic implementation of the validated EQ-5D-3L questionnaire. This is linked with an outcome measure of the study and is reported once per month. |
| R3 | Satisfaction with care reporting | Reporting of the satisfaction with received care using an electronic implementation of the validated PATSAT-C33 questionnaire. This is linked with an outcome measure of the study and is reported once per month. |
| R4 | Disease impact on family scale reporting | Reporting of the impact of the disease on the family using an electronic implementation of the validated Impact on Family Scale questionnaire. This is linked with an outcome measure of the study and is reported once per month. |
| R5 | Patient Symptom Reporting | Proxy reporting of symptoms for the patient using an electronic adaption of the SSPedi/Mini-SSPedi questionnaire. Usually patients will be asked to answer questions with regard to symptoms directly in the gamified app. |

QoL, quality of life.

**Table 4** The functionalities of the MyPal web-based app offered to the HCPs

| | Functionality | Description |
|---|---|---|
| H1 | Incoming information summary | A central page of the web-based app displays in a summarised form the incoming patient information that has not been reviewed yet. Incoming information is automatically prioritised in the system using custom algorithms whereby pieces of incoming information are assigned the highest priority and placed at the top of the list. Whenever an item is reviewed in full, it is removed from the list. |
| H2 | Individual data dashboard | A page that presents all the information collected for a given patient since the start of their participation in the study, using a dashboard with modern visualisations (see figure 4). The information includes primarily the patient's responses to the symptom questionnaire along with additional clinical information (eg, treatment, age group, diagnosis group, etc.). |
| H3 | Symptom Status over Time | This functionality and its visualisation are linked to the individual data dashboard. HCPs can document interim reports on current or past symptoms at different points in time as well as corresponding measures taken via selection of pre-defined choices. Additionally, the current or retrospective status of the patient's disease as well as the goal of treatment can be updated considering the whole course of the study. |
| H4 | Aggregated data dashboard | A page that presents aggregated and summarised information coming from all patients that participate in the study (descriptive statistics such a min, max, average and percentiles) using an analytics dashboard with modern visualisations. |
| H5 | HCP response log | A page that is used for logging potential responses (eg, referral to a specialist or prescription of medical examinations) of the HCP to the presented information of a specific patient. The HCP can log any actions taken after visiting individual data dashboard of a patient in a structured manner. |

MyPal web-based app are outlined in table 4. All of them can be accessed by the HCP at any time. In addition, the HCP can use the MyPal Carer app in order to complete questionnaires as a proxy for one of their patients.

### Data collection and analysis

All applicable national and European Union (EU) legislation, particularly the General Data Protection Regulation (EU 2016/679)[20] for the protection of individuals have been considered in the design of the study-related Information Technology (IT) infrastructure with regard to the confidential processing, collection and access to personal data. The technical deployment of the MyPal platform comprises of local server installations at each of the clinical study sites and one central server installation at the sponsor's site distinctly defining premises of data accessibility. The data security concept entails regular synchronisation of only anonymised data from local to central database. The apps being installed on mobile devices are protected by confidential credentials and store encrypted data only temporarily locally in case of absent internet connection until they are deleted from the mobile device subsequent to transmission to the respective databases.

### Sample size calculation

Assuming relatively acceptable values for the attrition rate (ie, 20%) and the missing data (ie, 30%), the sample size analysis concluded that 100 recruited paediatric patients providing one measure at enrolment (baseline) and six repeated measures (at Months 1, 2, 3, 4, 5, 6) are sufficient for the power of the intended statistical testing to be over 90% in all cases, given (1) a 0.05 significance level and (2) an effect size of 0.1; the employed value of the effect size was based on a priori knowledge of the domain, all power calculations were performed using the G*Power statistical analysis software.[21]

### Data analysis

The subsequent analysis and evaluation will serve to assess the feasibility of the MyPal digital healthcare platform

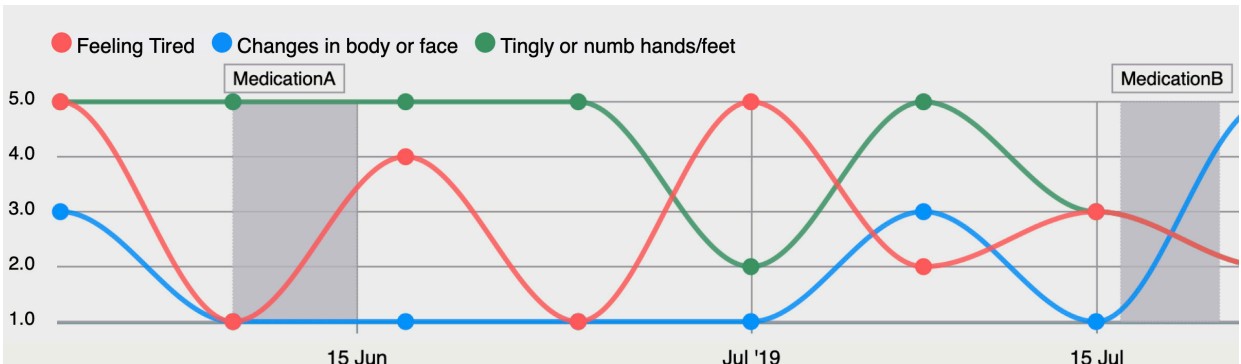

**Figure 4** Symptoms reported by study participants can be visualised in the MyPal web-based app for HCPs to be reviewed according to adjustable settings.

which uses ePROs to be ideally beneficial and supportive for the child and parents in palliative care.

Subgroup analysis of the outcome measures will be performed at baseline, month 3 and month 6 of the study using one-way, two-way and three-way analysis of variance (ANOVA) in order to detect potential differences between specific groups of participants. The level of significance for all statistical tests is set to a=0.05, in accordance with the power calculations. The grouping variables that will be assessed are (1) the clinical centre (origin), (2) the country of residence, (3) the age group, (4) the disease category (leukaemia, solid tumour, brain tumour), (5) the disease stage and (6) the starting point of the study in relation to the date of diagnosis. The latter point refers to recruited patients who have been under treatment for more than 6 months receiving care at the respective participating clinical site. These patients constitute a cohort for retrospective baseline data collection: the data collected from their completion of the ePROs at the beginning of their study enrolment will later be compared with outcome measure data derived from prospectively recruited patients whose treatment started within the last 6 months. Both groups will be followed likewise during the course of the study. Further data, like the daily number of steps (estimated by the smartphone/ tablet using the measurements of the built-in sensors) as well as the documentation of the symptom status over time, will be used to examine the feasibility to assess for correlations with reported symptoms and reported physical activity.

To evaluate the changes in outcome measures over time in the child cohort, repeated measures ANOVA will be applied on each measure (or a non-parametric equivalent). Post hoc analysis will be applied as appropriate. Since wording and layout of questionnaires were partly modified compared with the validated versions for electronic completion and gamification, another focus is (or would be) to assess the feasibility of such modifications to the original questionnaires. Repeated measures multivariate ANOVA (MANOVA) will be applied to the QoL of the children and the parents which serve as a pair of dependent variables.

To examine potential QoL differences between the participating sites, independent-samples t-tests and analysis of covariance (ANCOVA) models will be applied that will control for baseline criterion scores and potential confounders such as age group and sex which may be imbalanced between groups and associated with outcomes of interest.

The potential effect of the attrition is going to be assessed according to the methods introduced in Fewtrell *et al*,[22] for example, comparison of baseline characteristics of seen versus not seen at the follow-up participants to assess potential bias, sensitivity analysis, re-evaluation of the power of the study with the attained sample size. The impact of missing data in the attempted analysis is going to be evaluated based on the importance of the variables. Checks for data 'missing at random' and 'missing completely at random' are going to be performed.

Additional quantitative feedback will be gathered by a final evaluation of the MyPal digital health platform based on an appropriate usability one-time scale, that is, SUS for patients or parents, whereas HCPs' feedback will be collected via a designed web-based questionnaire at the end of the study.

## ETHICS AND DISSEMINATION

The integration of ePROs dealing with health issues into mobile applications raises ethical aspects which have been a crucial pillar in the implementation plan of this project under the monitoring of an internal ethics committee. Particular attention has been paid regarding data protection and the design of the MyPal-Child study protocol as it involves children being a vulnerable user group. The study protocol has obtained ethical approval from the ethics committee competent for the University of Saarland: Ärztekammer des Saarlandes (reference: Ha 23/20), the ethics committee of the Medical School Hannover (reference: 9095_BO_K_2020) as well as the ethics committee of the University of Brno (reference: 01–1 20 220/EK). The study results will be disseminated after study completion through publications in scientific journals, presentations at scientific conferences and congresses as well as a final report to the European Commission. Information on the MyPal project and newsletters, are publicly accessible via the project website (www.mypal-project.eu), Twitter (@ H2020MyPal) and Facebook (@MyPalProject). Based on study findings, concepts and strategies for the continuation of the MyPal platform with its tools and game as well as follow-up studies are considered.

### Concluding remarks

The foreseen advancement of the presented MyPal's patient-centred ePRO approach will offer a significant opportunity for children with cancer, their parents and healthcare providers to actively participate in the care process. By exploiting technological advances in ICT, MyPal aspires to contribute to bridge the gap between timely reporting and tracking of symptoms and the personalised actions performed by healthcare providers addressing the patient's needs which can vary across the disease course. The paradigm shift from passive patient reporting to active patient engagement could both enhance palliative and supportive care for children with cancer and improve coping with the disease.

**Author affiliations**
[1]Clinic of Pediatric Oncology and Hematology, Saarland University Hospital and Saarland University Faculty of Medicine, Homburg, Saarland, Germany
[2]Institute of Applied Biosciences, Centre for Research and Technology-Hellas, Thessaloniki, Central Macedonia, Greece
[3]International Observatory on End of Life Care, Institute for Health Research, Lancaster University, Lancaster, UK
[4]Department of Public Health Policy, University of West Attica, Athens, Attica, Greece
[5]Clinic for Paediatric Oncology and Haematology, Hannover Medical School, Hannover, Niedersachsen, Germany

⁶International Children's Palliative Care Network, Bristol, UK
⁷Head Office, European Association for Palliative Care, Vilvoorde, Belgium
⁸Department of Onco-Haematology, Division of Experimental Oncology, Università Vita Salute San Raffaele, Milano, Italy
⁹Department of Pediatric Oncology, University Hospital Brno, Brno, Jihomoravský, Czech Republic

**Correction notice** This article has been corrected since it was first published. The name of the author Julie Link has been corrected.

**Acknowledgements** MyPal is coordinated by Dr Kostas Stamatopoulos of the Centre for Research and Technology Hellas, Greece. Other partners are: Fraunhofer Institute for Biomedical Engineering, Germany; Foundation for Research and Technology Hellas, Greece; International Observatory on End of Life Care, Lancaster University, UK; Central European Institute of Technology, Masaryk University, Czech Republic; Karolinska Institute, Sweden; Vita-Salute San Raffaele University, Italy; University Hospital of Heraklion, Greece; Hannover Medical School, Germany; University Hospital Brno, Czech Republic; Saarland University, Germany; Promotion Software GMBH, Germany; Atlantis Healthcare, UK; European Association for Palliative Care; International Children's Palliative Care Network; National School of Public Health, Greece. The authors are grateful to the clinicians at the clinical study sites who are essential to this study (patient recruitment, symptom assessment and management and data collection). This work is dedicated to Vassilis Koutkias.

**Contributors** MM, NG, CK, CM, AS, SP, LS, PL drafted the study protocol MM and CK contributed equally as main authors on the manuscript. NG, TG-P, SP, AS, JD, CM, KS, JL, LS, CP contributed to critical revisions of the study protocol and the manuscript. NG, AS and PL are the site chief investigators and take overall responsibility for all aspects of the study design, the protocol and the study conduct at the involved clinical study sites.

**Funding** MyPal: Fostering Palliative Care of Adults and Children with Cancer through Advanced Patient Reported Outcome Systems, is funded by the Horizon 2020 Framework Programme of the European Union under Grant Agreement Nr. 825872.

**Competing interests** None declared.

**Patient consent for publication** Not required.

**Provenance and peer review** Not commissioned; externally peer reviewed.

**ORCID iDs**
Marcel Meyerheim http://orcid.org/0000-0002-9294-9445
Lydia Scarfò http://orcid.org/0000-0002-0844-0989
Norbert Graf https://orcid.org/0000-0002-2248-323X

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
