## [Reviewer comments · BMJ Open]

ARTICLE DETAILS

TITLE (PROVISIONAL)	The MyPal-Child study protocol: an observational prospective clinical feasibility study of the MyPal ePRO-based early palliative care digital system in paediatric oncology patients
AUTHORS	Meyerheim, Marcel; Karamanidou, Christina; Payne, Sheila; Garani-Papadatos, Tina; Sander, Annette; Downing, Julia; Stamatopoulos, Kostas; Ling, J.; Payne, Cathy; Scarfò, Lydia; Lokaj, Petr; Maramis, Christos; Graf, Norbert

VERSION 1 – REVIEW

REVIEWER	Bhawna Sirohi Apollo Proton Cancer Centre, Chennai, India
REVIEW RETURNED	30-Nov-2020

GENERAL COMMENTS	The authors describe an observational prospective clinical feasibility study of the MyPal ePRO-based platform for early palliative care digital system in paediatric oncology patients. The study is well written and thought through. Results will add significantly to the knowledge base globally. Please add the duration /timer period for the study to be conducted in the paper.
--

REVIEWER	Katherine Heinze Christine E. Lynn College of Nursing, Florida Atlantic University, United States
REVIEW RETURNED	03-Dec-2020

GENERAL COMMENTS	Thank you for the opportunity to review your manuscript describing your innovative study. Your proposed study addresses a very pressing need to improve symptom reporting in the pediatric oncology population, while giving children a voice in their care. You have clearly given much thought and care to the design of this study, and I am eagerly awaiting your results! I have a few suggestions for the manuscript before it is published: In the strengths and limitations listed at the top of the paper, the demands on HCP's time is not listed as a limitation, which I believe is one of the main limitations of the study. According to the study protocol, an HCP will need to review reported data every 72 hours - not a small ask. I'd also be curious to hear a little more about the benefits that you feel your study will have by recruiting participants from Germany and the Czech Republic. Is there something in particular about the
--

	patient population in these two countries that you feel will add to the study? As a reader outside of Europe, it would help me to understand the cultural makeup of your expected study participants, and the benefit in using these specific study sites. You mention in the study protocol that you have altered some of the study measures to fit your purposes for the study, but in the statistical analyses I didn't see a plan to validate these measures. Have you considered doing CFA to verify the validity of your alterations? Finally, the app itself sounds very interesting, and I was wondering if you could add maybe one or two sentences explaining the rationale behind the game with the diver. I'm not familiar with this branch of science (as I'm assuming many of your readers won't be), and I think it would help the reader to have a small explanation regarding the theory behind including symptom measurement in a game format. Do you expect the answers to match the answers given in the symptom surveys? Or is there something about the game format that you think will yield different results? Or is the reason for the game that it is just a vehicle to ensure that the symptom reporting is completed? Thank you again for the pleasure of reading your manuscript. I wish you all the best as you complete your study.
--	--

VERSION 1 – AUTHOR RESPONSE

According to Dr Bhawna's suggestion, who has been the first reviewer, we added the duration of the study in the section Study Design. Participants are enrolled for 6 months, recruitment started in December 2020 with an estimated study completion in March 2022.

We agree with the second reviewer, Dr Heinze, upon her opinion on the demands on HCP's time. Thus, we appended it to the limitations and strengths list as well. We would also like to comment that the healthcare professionals involved in the MyPal-Child study had been involved in the design of the study and agreed upon such a commitment. This point had already been addressed in form of a secondary objective as described in the Section Objectives and Outcome.

Considering the second reviewer's question on the chosen countries for recruitment: the participating clinical centres in Germany and the Czech Republic were chosen solely on the basis of their longstanding expertise in the management of paediatric patients with solid cancers and haematological malignancies. Indeed, in the planned statistical analysis, we do not intend to compare the patient population in the context of cultural differences, as this is beyond the scope of this feasibility study and requires to be properly addressed with a larger cohort of participants. Nevertheless, we are interested and will be comparing clinical centres in light of the palliative care they are offering or they have offered.

Considering the second reviewer's comment on the study measures: the changes we are referring to pertain only to the digital adaptation of the measures, not to the content which has remained otherwise intact. We have contacted the authors or developers of each measure and sought their permission for this digital adaption. Some have agreed to a digital adaptation without raising an issue with regards to the new version's validity. In the case where validity concerns were raised, we refrained from describing the adapted version as a 'validated measure'. The validation of the electronic versions of these questionnaires is indeed a worthwhile goal for future studies in the

eHealth-related observational and interventional studies. Unfortunately, it is currently beyond the scope of the current study.

Considering the second reviewers' comment on the rationale behind the game, we have updated the content of the sections Study Design and MyPal Digital Health Platform to provide a more comprehensive description of the idea behind the game. The main reason why a game format was chosen for symptom measurement is that it made it possible to break up a long questionnaire into different smaller sections such that it is a less mundane, more playful task for the child to complete. In other words, as you correctly pointed out, the game serves as a vehicle to encourage to complete symptom assessment over a long time period. In difference to a usual, non-gamified symptom survey, the children might be more spontaneous in their answers and more motivated to complete their survey in game format.

VERSION 2 – REVIEW

REVIEWER	Katherine Heinze PhD RN Florida Atlantic University United States
REVIEW RETURNED	18-Feb-2021
GENERAL COMMENTS	I hope you are safe and well. Thank you for your thoughtful revisions to this manuscript. I wish you much success in your study and I will look forward to seeing the results when they are published.